# The Role of Greenery in Stress Reduction among City Residents during the COVID-19 Pandemic

**DOI:** 10.3390/ijerph20105832

**Published:** 2023-05-16

**Authors:** Lidia Mierzejewska, Kamila Sikorska-Podyma, Marta Szejnfeld, Magdalena Wdowicka, Bogusz Modrzewski, Ewa Lechowska

**Affiliations:** 1Department of Spatial Planning and Urban Design, Faculty of Human Geography and Planning, Adam Mickiewicz University, 61-712 Poznań, Poland; kamila.sikorska-podyma@amu.edu.pl (K.S.-P.); marta.szejnfeld@amu.edu.pl (M.S.); magdalena.wdowicka@amu.edu.pl (M.W.); b.modrzewski@amu.edu.pl (B.M.); 2Faculty of Economics and Sociology, University of Lodz, 90-136 Łódź, Poland; ewa.lechowska@eksoc.uni.lodz.pl

**Keywords:** stress, sources of stress, COVID-19, city resilience, re-construction, health, urban greenery, biophilia

## Abstract

Cities, as places of social interactions and human relationships, face new challenges, problems, and threats, which are sources of stress for residents. An additional cause of stress in recent years has been the COVID-19 pandemic; it was urban dwellers who were most exposed to the virus and most affected by it. Chronic stress has led to the serious erosion of physical health and psychophysical well-being among urban dwellers, and so there is a need to seek new solutions in terms of building the resilience of cities and their residents to stress. This study aims to verify the hypothesis that greenery reduced the level of stress among urban dwellers during the pandemic. The verification of this hypothesis was achieved based on a literature analysis and the results of geo-questionnaire studies conducted involving 651 residents of Poznan—among the largest of Polish cities, where the share of green areas in the spatial structure is more than 30%. According to the analysis, the interviewees experienced above-average stress levels that went up during the pandemic, and the source was not so much the virus but the restrictions imposed. Green areas and outdoor activities helped in reducing this stress (being surrounded by and looking at greenery, garden work, or plant cultivation). Residents perceive a post-pandemic city as one that is more green, in which priority is given to unmanaged green areas. It has also been pointed out that a response to the reported need for urban re-construction towards stress resilience may be a biophilic city.

## 1. Introduction

We all live in a world full of stress. However, the number of potential stress-inducing factors is greater in cities [1]. Studies have shown that urban dwellers are increasingly more often exposed to stress and are also more vulnerable to stress than rural inhabitants. Moreover, city residents demonstrate a stronger brain reaction to stress, usually experiencing more severe cognitive impairment, anxiety, fear, and depression [2]. These effects seem independent of age, gender, overall physical health, civil status and level of wealth, but increase with city size [2]. Large cities, due to their complexity, are therefore more stressful than small towns [1].

However, the degree of stress experienced by a given individual depends on many elements. These elements, in turn, depend on the socio-economic development of particular countries or regions, and the resultant challenges they wrestle with (e.g., poverty, social polarization, bad hygiene conditions, a low level of safety, and climate change), and also on where and how one lives in the city. City life can be stressful when there is lack of sufficient living space, safety or stable economic conditions. The level of stress increases with the anticipation of unfavorable situations and concern about not having adequate resources to adapt to new circumstances. This leads to using energy for coping with stressful incidents, making the body more vulnerable to diseases [1,2,3]. 

Recently, the COVID-19 pandemic has become an acute stressor for urban dwellers, leaving its particular mark in cities. The virus causing the pandemic appeared unexpectedly, spreading dynamically as a result of globalization and international transport [4]. The pandemic affected all areas of life: family, work, and relationships. It limited everyday activity, making it impossible to sustain previous lifestyles [5,6,7,8,9,10]. 

As the cumulated effect of chronic stress has a profound impact on physical and mental health [6,11], physicians advocate for the need to reduce stress [12]. The World Health Organization (WHO) has recognized stress as among the main health challenges of the 21st century, and depression as a worldwide problem. However, the challenge may be faced by taking joint measures in the area of life sciences, social sciences, as well as city planning and architecture. As urbanization at a global scale is inevitable, there is an urgent need to improve knowledge of threatening—and also health-protection—elements of life in the city. Their identification of these elements is a starting point for the development of effective ways of limiting the degree to which people are exposed to stressors, in order to reduce their vulnerability to stress and the risk of health problems following stress during and after the pandemic [2,6,13,14,15]. Resilience can be understood as a product of the risk that a stressor will occur, the vulnerability to stress, which is the combination of susceptibility and exposure to stress as well as the duration of exposure [16]. For a long time, this concept has been associated primarily with the ability of a system to recover from a disturbance or the changes being implemented [17,18]. Regarding social systems, the process of building resilience emphasizes the importance of unique processes such as adaptation and transformation, which allow societies to respond to various types of threats [19,20]. Resilience can be defined in terms of a specific threat or a related set of threats [21]. In this publication, we examine resilience to a stressor such as a pandemic, with full awareness of the synergistic effects of different types of stressors in a city.

### 1.1. Sources of Stress in Cities

Stress can be described as an unspecific, physiological and psychological reaction to perceived risks to our physical, mental, and social integrity. Stress is a physical or mental tension appearing as a response to every stimulus recognized as potentially threatening to physical or emotional health. Anything that can result in death, destruction, or devaluation is a threat and a source of stress at the same time [1].

City residents have to wrestle with numerous stress-inducing elements, such as stimulus overload, continuous changes, congestion, noise, pollution, problems related to the functioning of public transport, cultural differences, and homelessness. This is additionally accompanied by interactions with a large number of people, fear of public opinion, anonymity, isolation, and loneliness [1,2]. Additionally, competitiveness and distrust in the workplace, growing and unpredictable requirements of employers, ever more pronounced materialism, consumerism, and social status keep urban dwellers in a state of uncertainty and anxiety about the stability of their work and means of livelihood. People often do not realize that these may be basic sources of their stress. Social stress, as an indirect consequence of high population density in cities, seems to be a leader among other stress-inducing elements, affecting the mental health of city residents [1,2]. Examples of sources of stress in the city are shown in Table 1.

### 1.2. The Effects of Stress

The unfavorable effects of stress, especially long-lasting effects, concern both the physical and mental spheres. In terms of the physical sphere, stress may increase the risk of cardiovascular diseases [22], type 2 diabetes and cancer [23], and is also a causal factor for obesity and premature aging [2], as well as neurodegradative diseases, e.g., dementia, Parkinson’s disease and Alzheimer’s disease [23,24]. Exposure to stress, especially long-lasting stress, may additionally weaken the immune response, increasing vulnerability to various types of illnesses [25]. As for the mental sphere, stress may lead to anxiety, depression, schizophrenia, paranoia, etc., and even suicide [1,2].

### 1.3. Stress in the City during the Pandemic

The stress level of urban residents increased from early 2020 as result of the COVID-19 pandemic. The emergence of this sudden stressor with its unimaginable consequences activated an avalanche of events leading to progressive destruction, which could not be immediately stopped, counteracted, or avoided [4].

The most stress-inducing socio-psychological elements in the urban environment during the pandemic were the economic slowdown, real and foreseen financial losses, isolation accompanying quarantine, supply shortages, uncertainty, loneliness, fear of infection, stigma associated with contagion or its risk, resulting in social phobia, and boredom [26]. Uncertainty and isolation were particularly problematic.

Uncertainty surrounding the pandemic was a completely new phenomenon for many populations. The lack of warning of an approaching threat made it impossible to prepare for the upcoming situation and impeded preliminary, healthy adaptation [3,26]. Ubiquitous uncertainty around the pandemic hindered future planning, thus becoming a source of new stressors [27].

Social isolation results from a lack of important relations and social contact with others [28,29]. In normal circumstances (outside the pandemic period), isolation only affects some groups of city dwellers, such as widowed persons and the elderly, who are particularly exposed to loneliness. During the COVID-19 pandemic, these threats were exacerbated because of imposed restrictions (limitations to movement, contacts with family and friends, etc.), quarantine and necessary self-isolation, affecting other groups of residents (e.g., the young from single households) [26,30,31,32].

The emergence of the pandemic resulted in an array of negative emotions (e.g., sorrow, fear, anger, guilt and regret, as well as the feeling of losing control over one’s life, which may be symptoms of mental disorders) [33]. As a result, a new disease was identified, namely pandemic acute stress disorder. This was a reaction to traumatic stress resulting from the response to a sense of threat accompanied by fear, uncertainty, concern, depression or anxiety disorders [4,33].

Due to the numerous sources of stress in the city and the increased exposure to chronic stress, the priority became taking measures to help residents cope with stress during the pandemic and afterwards, strengthening the resilience of cities [6]. Strengthening resilience to stress and protection against its negative consequences for individuals will have a positive effect on the resilience of communities, helping them to sustain and even develop in difficult circumstances [30].

Adopting urban health programs [34] that emphasize the role of proper spatial development and natural urban resources, which affect human health, may help build resilience to stress among city dwellers [35,36,37,38,39,40]. 

### 1.4. The Role of Greenery in Reducing Stress among Urban Dwellers

Over the centuries, writers, philosophers and naturalists have emphasized the benefits of the natural environment to human health, happiness and well-being, but only relatively recently have they started to explore and quantitatively determine complex relations between human health and nature. This research shows that the natural world contains chemical and biological elements that have a positive physiological and psychological effect on the functioning of the immune system [25]. Therefore, contact with the natural environment is conducive to strengthening resilience.

Green areas are associated with many pro-health benefits. What has been proved is that there is a positive correlation between the availability of green or natural environments and the commonly accepted general state of health [41,42], mental health [42,43,44,45], longevity [46], physical health [47,48] and social health [41,42,49,50,51]. Higher exposure of residents to greenery results in reduced mortality [35]. This means that access to green areas near to one’s place of residence increases the chances of survival [46,52,53]. There has also been epidemiological research at the national level, involving cases and experimental studies, on the relationship between greenery and health [1].

What has also been proved is the relationship between greenery and stress and anxiety reduction [42,43,44,54]. Both reported and objective (measured by the level of cortisol) stress levels fall with an increase in the amount of green space in a local community. This happens regardless of the quality of greenery or the perception of such spaces by users [1]. A positive effect on the mood and self-assessment is observed already after five-minute exposure to green areas, independently of the intensity of activity [55].

It is not only visiting green spaces that reduces stress, but also looking at green spaces [54]. Therefore, contact with the natural environment is conducive to mental renewal [56], mood improvement [44,55,57], and concentration improvement [44,45].

Japanese scientists have found that being surrounded by nature may be among the most powerful relaxation agents. The research shows that forest environments may contribute to a lower concentration of cortisol, reduce the heart rate and blood pressure, stimulate parasympathetic nervous system activity and decrease sympathetic nerve activity compared to urban environments [58,59]. As a result, ‘forest bathing’ (Shinrin-Yoku: taking in the forest atmosphere) has become a popular way of relaxing in Japan and other countries [60].

A mechanism of those relationships has been explained, e.g., in Ulrich’s stress reduction theory (SRT) [54,61]. The theory is based on empirical research which showed the immediate positive reactions of individuals to natural conditions [60]. A view of nature helped stressed people by reducing blood pressure [44,54], muscle tension [54] and heart rate [54] within a few minutes [60]. Some places, such as those with abundant vegetation, calmly rippling or slowly moving water, or greenery modelled on the savannah, were also found to be increase the chances of post-stress recovery [60]. According to Ulrich, organisms that have developed an immediate regeneration ability over millions of years of evolution have a better chance of survival (especially in areas that provide safety and access to food, and by remaining “mentally alert” after surviving stressful situations).

In the context of stress, greenery serves preventive (enhancing physiological resilience), reducing (reduction in the stress level among urban dwellers) and regenerative (effective and fast post-stress recovery) functions (Figure 1).

The transformation of cities towards ensuring that residents have access to appropriately landscaped green spaces (enhancing their areas, accurate smart spatial layout, generic structure, etc.) [13,62,63] fits in with long-term measures of strategic importance (coping strategies) for city resilience, determined by the term re-construction.

### 1.5. Hypothesis and Research Objectives

The research procedure in this work is basically aimed at verifying the hypothesis that greenery reduced the level of stress among urban dwellers during the pandemic (also in a post-pandemic city). On the other hand, the specific objectives relate to the determination of: (1) the level of stress among city residents during the pandemic, (2) the sources of this stress, (3) the effects of pandemic stress, (4) elements of the city’s spatial structure that increased and reduced stress among residents during the pandemic, (5) the factors that reduce stress among residents, as well as—the role of greenery in mitigating stress. By the term greenery, we mean all areas in the city covered with vegetation (regardless of the type of ownership). We also point out the concept of a biophilic city, which can be treated as a model of structural transformation to improve the resilience of city dwellers to stress and build a stress-resilient city. The basis for achieving the formulated research objectives and verification of the hypothesis was the results of a geo-survey, conducted among the residents of one of Poland’s largest cities—Poznan.

The literature is dominated by recreational [64,65,66,67,68,69] and ecological [70,71,72,73,74,75] approaches to urban greenery. In our article, on the other hand, we pay special attention to the importance of transforming post-COVID-19 cities in the direction of increasing the availability of greenery for residents to improve their health, both mental and physical health, thus filling the existing research gap. What we point out are the main sources of stress for urban residents during a pandemic, the spatial distribution of stressors, and ways to reduce pandemic stress. We also emphasize the preventive, reductive and regenerative role of greenery, the appropriate shaping of which can become an important element in the process of accelerating the “recovery” of cities and their residents from stress in the post-pandemic era and building their resistance to stress.

The research conducted relates to the Sustainable Development Goals published by the United Nations [76], among which Goal 11 applies to “making cities and human settlements inclusive, safe, resilient and sustainable” [76]. The word ‘safe’ refers to the notion of health, which is determined by the World Health Organization (WHO) as “a state of complete physical, mental and social well-being and not merely the absence of disease or infirmity” [77].

## 2. Materials and Methods

The introduction contains the hypothesis stating that greenery reduces the stress level of urban residents in pandemic conditions. The verification of this hypothesis required evidence that the pandemic was a stress-inducing factor for city dwellers, and also that greenery made it possible to reduce the accompanying tension. To achieve this, a number of research methods were used, including the analysis of literature, documents and source materials collected. The most important was, however, the field research studies (geo-questionnaire) and statistical methods for developing the results. 

The geo-survey is based on a traditional questionnaire, which has been supplemented with questions related to identifiable aspects of space. The interactive map used in the geo-survey allows respondents to indicate, characterize, and, as appropriate, evaluate elements of space (points, lines, and areas, as used in GIS software (Esri, Redlands, CA, USA)) [78]. Achieving the desired number of geo-survey respondents involves selecting the right methods to promote the consultation to residents. The most effective methods of recruitment include news in the local media, flyers delivered directly to mailboxes, and information on social media [79].

The pilot geo-questionnaire studies were carried out in the period July–September 2022 involving the residents of Poznań city, among the largest of Polish cities, with a population of 543,347 in 2022 [80], a metropolis with regional coverage. It is a city characterized by a high share of green areas as a percentage of total land area (as much as 30.5% city area in 2019, not including arable land, lakes and rivers, which implies approximately 100 sq. m. of greenery per capita) [81], which in this category places the city in the second position among all Polish metropolises [82]. The studies were aimed, among other things, determining a subjectively perceived level of pandemic-caused stress by residents, the sources of that stress, ways that helped in reducing stress during the pandemic, and also places in a spatial urban structure which were able to increase or reduce stress.

The geo-survey was conducted in an online form and was addressed to all the residents of Poznań over the age of 18. Since the survey is part of a larger research project, information about the geo-survey was posted on the project website, disseminated in cooperation with local authorities (the website of the Poznań City Hall and the office’s FB), social media, and promoted in local media (local radio, newspapers, the website of Adam Mickiewicz University, Poznań).

The geo-questionnaire studies covered 651 people altogether, including 430 women, 202 men and 19 individuals who identify their gender differently (cf. Table 2). Due to the fact that not all geo-questionnaire questions were obligatory, the number of answers to particular questions is diversified. The interviewees were mostly young residents of Poznań (18–25 years old), which made up 43.9% of respondents and mature people (36–60 years old)—29.8%. The vast majority of respondents had higher (62.5%) and secondary education (35.3%).

The present survey is based on 9 geo-survey questions (not counting the questions included in the metric), 7 of which were of closed character. The content of the questions and the type of answers are presented in the form of a Appendix A. Two of the questions asked involved marking on maps specific places that increase and reduce respondents’ stress in the city, and specifying the type of place indicated (e.g., park, restaurant, gym).

Among statistical analysis methods, apart from simple measures, such as the mean and standard deviation, use was made of correlation analysis for research into relationships between the level of stress reported by interviewees and the perceived role of greenery in its reduction and the characteristics of the respondents.

In order to determine the correlation between the variables obtained in the questionnaire, a non-parametric Chi^2^ test was conducted. It can be used to analyze conformity of both measurable and non-measurable features. A chi-squared test compares the observed values and the expected ones. To check statistical significance of correlations, one should compare the calculated chi-squared statistics with theoretical statistics. The test itself makes it possible to determine if a given correlation exists or not. On the other hand, the *p*-value indicates whether the identified correlation is less or more probable. The lower the *p*-value, the greater probability that the correlation is true. Therefore, one may assume that the lower the *p*-value, the stronger the identified correlation. In the research conducted, the significance level adopted was 0.05.

Additionally, in order to complete the analyses, what was also determined for the correlation between the age and the stress level was Spearman’s rank correlation coefficient applied to describe the strength of the correlation of two features (both quantitative and qualitative), which may be organized by giving them ascending or descending ranks.

## 3. Questionnaire Results

As was mentioned earlier, urban dwellers are particularly severely exposed to stress today. This has been confirmed by the results of the geo-questionnaire pilot research conducted among the residents of Poznań city in 2022 as part of the scientific project “Stress-resilient city during the pandemic (COVID-19)”. 

During the COVID-19 pandemic, the respondents assessed their stress level on a 0–10 scale (0—lack of stress; 10—the highest stress level) at 5.47 on average. Stress was not experienced at all by 5.30% of the respondents, whereas 6.99% individuals suffered from highest level of stress. 

The stress level experienced by the respondents depends primarily on sex/gender (women are more stressed than men) and education (the higher the education the higher the stress level) (Table 3). However, it does not depend on occupational activity and the interviewees’ age. Still, in the second case, the issue is arguable. Indeed, Chi^2^ tests have not confirmed the relationship between the age and the stress level of the interviewees but Spearman’s rank correlation coefficient has demonstrated a positive, statistically significant correlation (rs = 0.1122, *p* = 0.015).

Over 40% of the respondents reported an increase in the stress level experienced during the pandemic, whereas 35.81% noticed a decrease, and 23.52% of the respondents stated that stress was maintained at the pre-pandemic level. 

The change in the level of experienced stress is related to almost all demographic–economic features of the respondents (except for education), primarily to gender (a smaller change in its level is observed for men) (Table 4).

It is worth emphasizing that there is a strong correlation between the current level of and change in stress among respondents (Chi^2^ test = 79.0387, *p* = 0.0000000000001). The existing high stress level of the interviewees results from probably its increase during the pandemic.

What should be noted is that the most stressful element for the residents during the pandemic was not the virus itself, but a series of the resultant restrictions imposed in urban spaces. While the presence of the virus was highly disturbing for a mere 37.71% of the interviewees, as much as 78.60% of them experienced stress related to the fear of rising prices, 69.70% because of difficulties in accessing health services, 64.83% in relation to the restrictions on movement and leaving apartments, 56.36% feared losing income and 54.24% felt threatened because of travel difficulties (Table 4).

The residents of Poznań mainly indicate psychophysical aspects of their health deteriorating resulting from the excessive stress experienced, such as weakness and general reluctance to act (72.78%), problems with concentration and memory (67.37%), sleeping disorders (52.54%), anxiety attacks (42.16%), growing family conflicts (41.10%) and compulsive eating (36.65%) (Figure 2).

The respondents’ stress was exacerbated primarily by staying in closed public spaces within the city, including in health care institutions (indicated by 61.86% of the respondents), large shopping centers (50.00%), railway stations and public transport stops (47.03%), public facilities—offices (47.03%) and churches (42.16%). In contrast, open spaces, in particular green spaces (parks, small green areas, allotments), contribute to reducing stress among city dwellers, which was indicated by 66.95% of the respondents, and outdoor recreational areas (55.72%) (Figure 3).

An important role of greenery in reducing stress is illustrated in Figure 4, which shows specific places within the city exacerbating and reducing the respondents’ stress. It is easy to notice that places where stress is reduced are concentrated mainly around green areas (e.g., the Citadel Park, the Adam Wodziczko Park, areas around lakes and the Wartariver), whereas those exacerbating stress are mainly public transport stops, railway stations (including the Poznań Central railway station) and shopping centers (e.g., Malta, Posnania, Plaza).

The following activities help to reduce stress to the greatest extent: being surrounded by green areas (94.70%), a view of greenery (92.16%), outdoor activity (88.77%), garden work and plant cultivation (64.19%). On the other hand, the COVID-19 vaccination reduces stress to a small degree, which has been indicated by a mere 31.78% of the respondents (Figure 5). However, there is also a number of people whose stress during the pandemic was reduced by contact with family, as well as with pets, pursuing hobbies, watching movies or using social media.

Interestingly, the stress-reducing role of greenery was primarily indicated by those individuals who reported a decrease in their stress level and described it in the questionnaire as currently relatively low (Table 5). Therefore, one may assume that a decrease in the subjective stress level of the interviewed city residents may result from a positive influence of green areas.

What requires an explanation is that the stress level of the respondents (also the changes in it) is more related to a positive impact of greenery than to socio-economic features of the respondents, which is shown by the obtained values p for Chi^2^ tests (Table 4 and Table 5).

It is worth mentioning, given the great importance attached by Poznań residents to green spaces as an element mitigating stress, that 17.5% of the respondents indicated scarcity of green areas at a nearby place of their residence during the pandemic. Simultaneously, they clearly point to the need for urban development towards increasing the proportion of green areas (on the scale from 0—built-up areas, to 10—green spaces, the average assessment was 8.86), with emphasis on creating unmanaged green areas that are biodiverse complexes, as close as possible to natural ones (on the scale 0—unmanaged greenery, to 10—organized green spaces, the average assessment was 3.86).

## 4. Discussion of the Results

The results of the research conducted among Poznań residents confirm the theses developed in the literature that urban dwellers perceive their stress level as above-average [1,2] and that the COVID-19 pandemic exacerbated this stress [4,26,27,30,31,32]). The respondents felt pandemic-related stress despite the fact that almost all pandemic constraints have been removed in Poland since March 2022, making it possible for some respondents to experience a decrease in pandemic stress. 

The symptoms of pandemic-related stress among the respondents are mainly a weakness and general reluctance to act, problems with concentration and memory, sleeping disorders, and, albeit to a smaller extent, anxiety disorders. 

It is worth emphasizing that the restrictions imposed were more stressful for the interviewees than the virus itself. Particular tensions were related to concerns about rising prices, restricted access to health services and constraints on leaving apartments and movement.

The results are also confirmed by conclusions drawn from many publications about a beneficial impact of greenery and outdoor recreation on stress levels [42,43,44,54]. Being surrounded by greenery helped almost all interviewees to reduce their stress levels (nearly 95%). What turned out important, however, was also looking at greenery itself, which confirms Urlich’s assumptions [54], and also garden work and plant cultivation. 

The hypothesis put forward in the introduction has been proved by the research results obtained. The results also reveal the need to re-construct the spatial structure of post-pandemic cities towards increasing the share of green spaces [83,84,85] and, interestingly enough, unmanaged greenery is particularly desirable. Researchers seem to notice benefits from biologically active areas left in their natural form, described, in the shinrin yoku conception, which can be developed in a naturalistic way.

The need to properly develop green spaces is an important message for urban planners when designing new housing estates, regenerating the existing urban structure or during consultations about land-use priorities [1]. The studies carried out by Kuo [86] show that the presence of biologically active areas alone near houses, schools, hospitals and workplaces appears to be advantageous. The residents of public facilities situated in the vicinity of vegetation may cope with stress more effectively compared to those living in buildings surrounded by concrete [86]. Moreover, greenery around heavy traffic roads may lower annoying noise levels [87,88], and flora may increase privacy and hide unsightly parts of construction [89]. A response to this need, observed in the literature and proposed by the respondents, may be introducing ‘nature’ to the urban environment based on the biophilia conception.

In the biophilic design, attention is paid to increasing the area of urban greenery (also that is ‘unmanaged’ but biodiverse and home to numerous species instead), and by using all possibilities for implanting green development on various scales and various urban surfaces, including unused car parks, walls (inside and outside) of buildings, green roofs, individual and social gardens, etc. [90]. What is significant in this approach, however, is not only increasing the area of ‘nature in the city’ alone, but also making it possible for residents to have active, physical contact with it every day. The positive effects of a biophilic influence are related to the imitation of the effects of staying around real, natural phenomena in an artificial (architectural) environment, and also to modelling the principle of environmental complexity, which implies the importance of not only greenery, but the role of architectural geometry as well in mitigating stress [91,92]. The array of biophilic stimuli embraces visual (natural view, landscape) and auditory impacts (e.g., white noise), as well as tactile (e.g., material textures) or olfactory ones (such as in Shinrin-Yoku). The biophilic benefits are observed in many dimensions—cognitive (better concentration, engagement and memory), emotional, regenerative and in relation to stress reduction and higher productivity [93]. In this context, it would be advantageous to use biophilic principles for (re)designing closed and open public spaces that exacerbate stress among urban residents (public transport stops, railways, hospitals, government offices, etc.) as well as the introduction of greenery in various forms wherever possible. Such action is desirable, especially in highly urbanized areas, where the share of biologically active lands is small.

Urban greenery, which to some extent can mitigate the effects of exposure to urban stress, should therefore be treated as a means to the end of the revitalization of public spaces, and thus the economics and biological development prospects of city residents. Greenery itself, however, will not fully substitute the essential biophilic aspect of cities, which should be their health-promoting, diverse and rich environment, enabling the development of residents in all aspects, including psychological, physical and biological. 

### Limitations of This Study

Our research, as any other, has its limitations. It was conducted in only one Polish city of a certain size, with a limited sample of residents. Additionally, although the residents of Poznań are a relatively homogeneous group (minor differences in nationality, ethnicity, culture, religion, etc.), and there is practically no sub-standard housing in the city space, generalization of the results of this study should be made with great caution. This is because the course of the pandemic in each city is individualized [94], as is the extent of pandemic restrictions imposed on residents and the frequency of changes made in reference to subsequent waves of the pandemic. The results of this study were also undoubtedly influenced by two conditions: (1) the relatively rapid, compared to other countries, lifting of pandemic restrictions, resulting from (2) the appearance in Poland of a new stressor, which is the war going on just across the Polish border with its various consequences (a wave of refugees, the organization of aid for the residents of Ukraine, the increase in the price of fuel and other products, etc.).

## 5. Conclusions

Living in a city brings many benefits, but it is also a source of stress for residents. In this article, we pointed out the sources of this stress, which recently included the COVID-19 pandemic. We showed that due to the harm of stress to the physical and mental health of residents, it is necessary to seek ways to reduce this stress, and urban greenery has a large role to play.

As a result of questionnaire surveys conducted with the residents of Poznań, we determined the level and sources of stress for residents during the pandemic. We obtained rather surprising results, from which we found that greater sources of stress for the respondents were the restrictions imposed than the virus itself. We also pointed out elements of the city’s spatial structure and activities that increase and reduce stress. We confirmed the hypothesis formulated in the literature that urban greenery has a stress-reducing function for residents, in addition to a preventive (supporting the immune system) and regenerative (accelerates recovery) function. Indeed, it turned out that greenery reduces stress among residents more than vaccination against COVID-19. Respondents reported the need to shape cities with more greenery, specifically with a large share of unmanaged green areas. This implies an urgent need to transform the spatial structure of cities to that which creates healthy living conditions for residents and increases their resilience to potential future pandemics, paying particular attention to proper design of green spaces.

### 5.1. General Recommendations

Based on the results of the conducted research, some recommendations for both city authorities and experts (planners, urban planners, architects, etc.) responsible for development planning and urban design may be offered. These recommendations should be considered while planning the development of healthy, stress-resilient centers:(1)Post-pandemic cities require the re-construction of their spatio-functional structure in such a way that they could provide residents with a healthy living environment, free from various harmful and stress-inducing elements, or at least minimize the influence of those elements, contributing simultaneously to reducing stress levels.(2)This re-construction should aim primarily at increasing the volume of urban greenery, especially that which is natural, with its great complexity and biodiversity.(3)What is important is not only an increase in the proportion of green space in the urban structure and its concentration in selected points or places, but its dispersion over the areas performing different functions and the creation of whole, diverse systems of biologically active areas. Such a practice will allow equitable access to green space for all city residents, creating conditions for stress reduction.(4)Both open and closed public spaces require transformation, including primarily those exacerbating stress among residents (public transport stops, railway stations, shopping arcades, health care institutions, etc.).(5)It is important to enable residents to interact with nature as often as possible on a daily basis (at least a five-minute contact with nature in a local living environment, both near their place of residence and their workplace).(6)The conception of a biophilic city may serve as a model for such a perceived re-construction of post-pandemic cities.(7)Greenery in cities may adopt various forms; it may cover not only the surface, but also roofs, or building facades.

### 5.2. Recommendations for the City of Poznan

As the results of the geo-survey showed, the main sources of stressors in Poznań during the pandemic were not virusogenic, but psychological in nature, resulting in psychosomatic symptoms that were troublesome in the daily individual and social functioning of city residents during and after the COVID-19 pandemic. Individual psychological problems, however, can accumulate into deterioration in the overall social health of city residents, which in turn will affect their productivity and the cost of their recovery and regeneration. This will raise challenges for reorganizing the nature of emergency and clinical (psychological and psychiatric) care sites, especially for residents who are alone, lack family support and do not have their own home gardens.

In addition to the challenge of redesigning health infrastructure sites, there is an urgent need to redefine the role of urban public spaces themselves, especially the functioning of the network of public transport points and hubs. It is necessary to abandon the approach to them as purely technical or representational spaces and seek health-promoting and salutogenic solutions, as it were, by design (intentionally), mitigating the stresses arising from the pandemic and post-pandemic nature of urban life.

It is also worth noting that while respondents cited being in greenery and working in gardens as reducing stress, there is no organized, planned form of either: (1) community gardens (allowing those without private gardens to conduct such activities), or other open areas for psycho-physical activity and horticulture for adults, especially in conjunction with health-promoting infrastructure; (2) intentionally designed downtown biologically active areas, (providing a full spectrum of biodiversity); (3) plans for an alternative system of zero-emission transportation corridors based on the city’s green infrastructure.

The main places in the spatial structure of the city of Poznań stressing residents during the COVID-19 pandemic (and beyond) are concentrated primarily in the inner city, characterized by a high degree of housing density and a small share of green space. Measures to develop greenery and, as a result, reduce stress on residents in this part of the city include
(1)Greening of internal courtyards in quarters of compact frontage buildings, allowing residents of all ages to interact with nature (community gardens, natural playgrounds, sensory gardens, etc.);(2)Elimination of some areas of paved city squares (e.g., Liberty Square) in favor of introducing greenery that promotes biodiversity;(3)Revitalization of traffic routes, streets, through the introduction of linear forms of greenery (green tracks, tree and shrubbery between individual lanes, etc.) as well as through the creation of green enclaves in the form of urban stops, made of natural materials and surrounded by greenery;(4)Transformation of former post-railroad areas, among others, into forms of “linear urban parks” (e.g., the area of the so-called “free tracks” near the Central Station);(5)Revitalization of existing watercourses, greening of their banks, resignation from “canalization” of smaller watercourses in favor of leaving them on the surface and making them available to residents,(6)Transformation of large surface parking lots into biologically active areas or supplementing these spaces with elements of green infrastructure, especially at health care facilities and public buildings;(7)Introduction of “green facades,” especially on the numerous, often windowless gables of buildings, and “greening of roofs” on technically compliant buildings;(8)Greening of open spaces around and within transportation hubs and large-format shopping centers (e.g., Main Railway Station with Avenida Gallery, Posnania shopping center), as well as the introduction of “green courtyards” inside such facilities.

The public demand for the realization of new green areas, and the planning challenge they may pose in the face of increasing levels of urban stress and its consequences, may be the subject of further in-depth study and survey research.

### 5.3. Future Research

Ensuring the sustainability of development processes requires building the resilience of cities and their residents to various types of threats (including pandemics). Related to this is the need to deepen research on possible ways to reduce their vulnerability to a given stressor. In order to shape stress-resilient cities, it is worth drawing on the experience gained during the slowly ending COVID-19 pandemic. In this context, the results presented in this article open up new fields of research for similar studies, conducted for other cities, for collections of cities, as well as for analyses of a comparative nature. 

Expanding research on the risks, sources and consequences of a given threat, as well as ways to reduce stress (not just pandemic stress) in a city, will help improve the quality of life of residents and build healthier, more livable and sustainable cities. It will also allow us to be better prepared for future epidemics and pandemics.

It is worth emphasizing at the same time that greenery, despite a direct influence on stress reduction (not only during a pandemic, but at all time), also performs other, important functions for the city and its residents. For instance, it improves their general health; limits the risk of civilizational diseases; reduces the level of nuisance, such as noise, air pollution, the effect of an urban heat island; prevents the negative effects of sudden heavy rainfall; improves the urban microclimate [84,85,89,95]. It is therefore also a response to the observed progressing climate change and also to ever more frequent heatwaves in cities.

## Figures and Tables

**Figure 1 ijerph-20-05832-f001:**
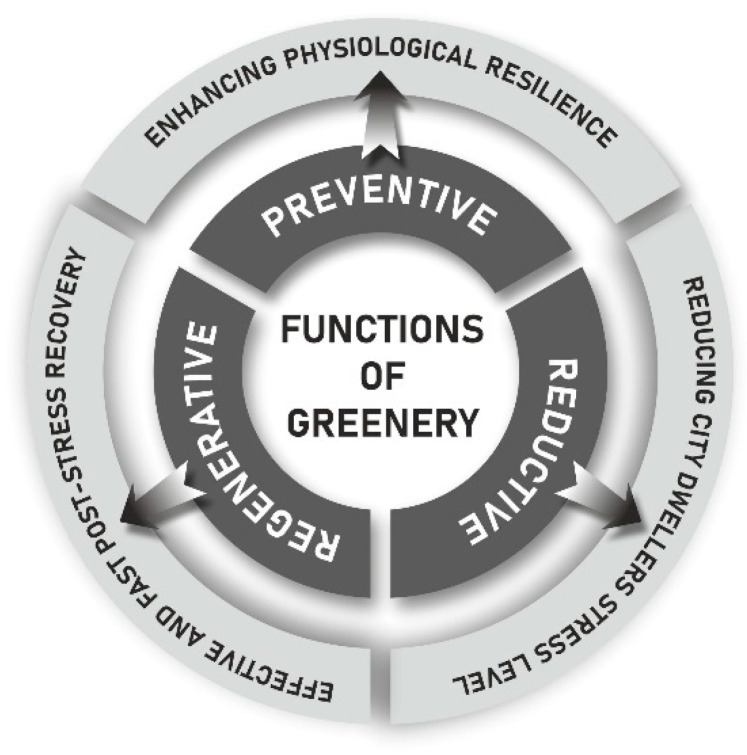
Functions of greenery in relation to stress urban dwellers. Source: own compilation.

**Figure 2 ijerph-20-05832-f002:**
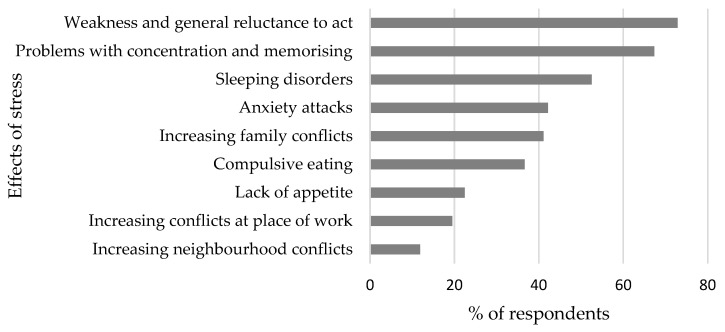
Effects of stress caused by the COVID-19 pandemic. Source: own study on the basis of questionnaire research.

**Figure 3 ijerph-20-05832-f003:**
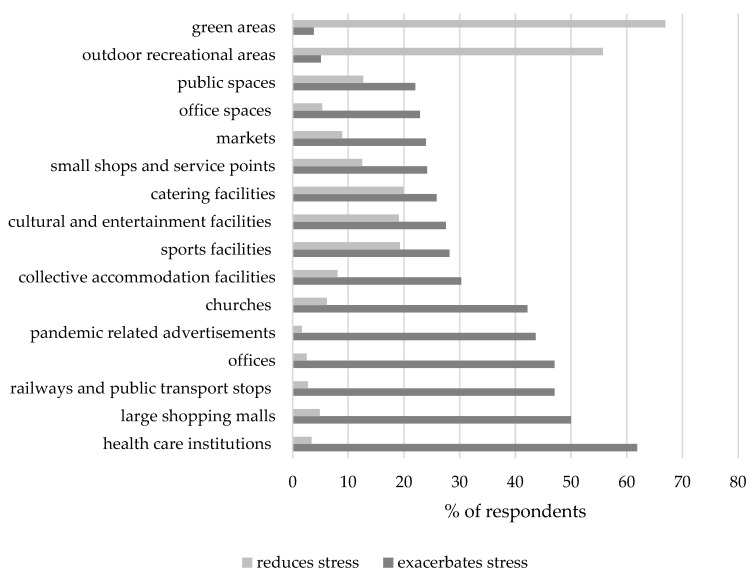
City spaces exacerbating and reducing stress during the COVID-19 pandemic. Source: own study on the basis of the questionnaire results.

**Figure 4 ijerph-20-05832-f004:**
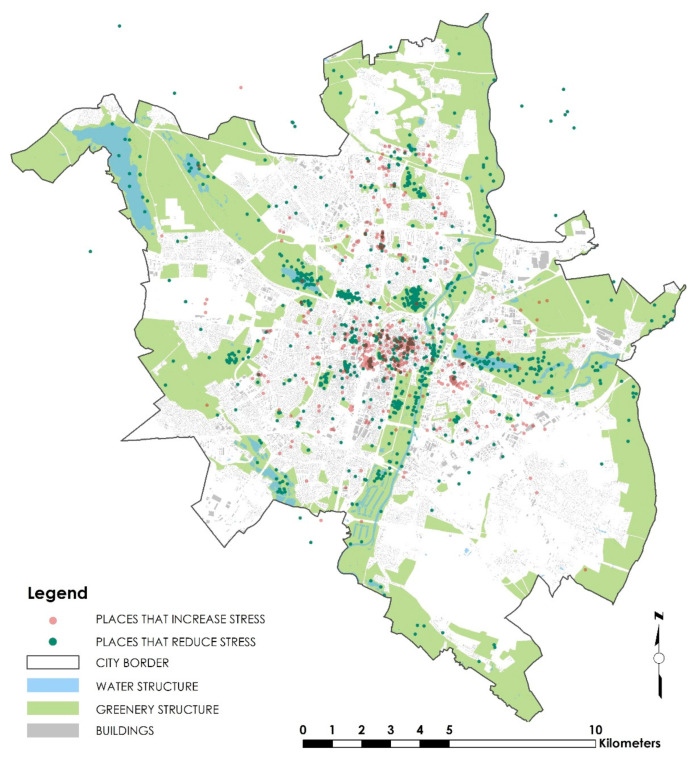
Places exacerbating and reducing stress among Poznań residents. Source: own study on the basis of the questionnaire results.

**Figure 5 ijerph-20-05832-f005:**
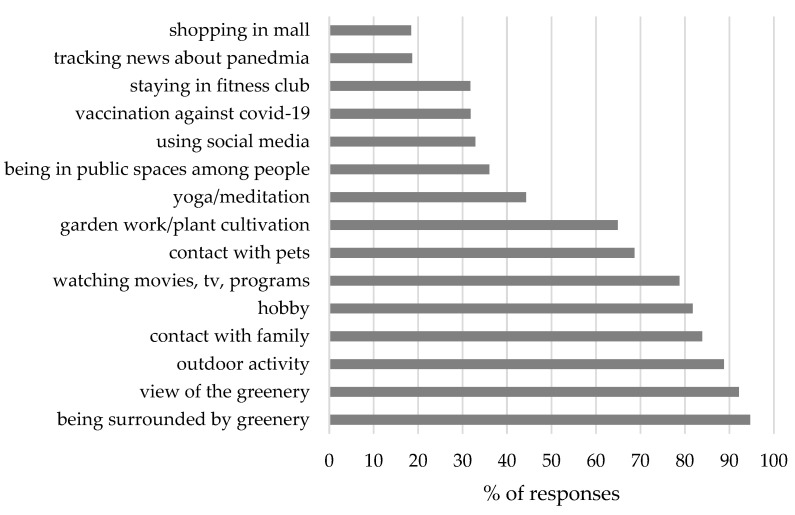
Elements reducing the stress level of the residents during the COVID-19 pandemic. Source: own study on the basis of the research results.

**Table 1 ijerph-20-05832-t001:** Examples of sources of stress in the city.

Group of Stressors	Examples
**Socio-psychological**	poverty; social disparities; disturbance of chronobiological rhythms; uncertainty *; reduced life control *; isolation *; stigmatisation *; unresponsive bureaucracy; impersonal treatment
**Environmental**	polluted air; noise; contamination with light; urban heat island
**Economic**	risk of losing job *; competition on labour market; pressure to increase productivity; being subjected to various work demands; unpredictable employers; price rise *
**Architecture and urban planning**	fast pace of life; stimulus overload (colors, sounds, people, information, events, etc.); urban system failures (causing, e.g., delays in reaching place of work/home); excessive advertising; restrictions on availability of goods and services *

* Sources of stress that had a powerful impact during the COVID-19 pandemic. Source: own study based on [1,6].

**Table 2 ijerph-20-05832-t002:** Respondents’ structure.

Details		No. of Respondents	Share of Respondents (%)
**Gender**	Female	430	66.05
	Male	202	31.03
	Other	19	2.92
**Age**	18–25	286	43.94
	26–35	150	23.04
	36–60	194	29.80
	61–70	16	2.45
	Over 70	5	0.77
**Education**	Higher	407	62.52
	Secondary	230	35.33
	Vocational	11	1.69
	Basic	3	0.46
**Occupational Activity**	Pupil/student	241	37.02
	Unemployed	16	2.45
	Public sector	156	23.97
	Private sector	175	26.88
	Own business	48	7.37
	Pensioner	15	2.31

Source: own study based on questionnaire research.

**Table 3 ijerph-20-05832-t003:** Correlation between demographic–economic features of urban dwellers and the subjective level of and change in stress (Chi^2^ test results and *p*-value).

Stress	Demographic–Economic Features			
	Age	Gender	Education	Occupational Activity
**Stress Level**	16.1724 (*p* = 0.4410)	13.5488 (*p* = 0.0089)	9.1062 (*p* = 0.0585)	5.8487 (*p* = 0.6642)
**Change in Stress Level**	14.2384 (*p* = 0.0271)	10.4780 (*p* = 0.0053)	0.6459 (*p* = 0.7240)	12.8041 (*p* = 0.0463)

Source: own study on the basis of questionnaire research.

**Table 4 ijerph-20-05832-t004:** Stressors during the COVID-19 pandemic.

	Stress Level (% of Respondents)		
Stressor	Low (0–3)	Medium (4–6)	High (7–10)
**Presence of COVID-19 virus**	31.78	30.51	37.71
**Uncertainty as to changes in restrictions**	21.61	21.61	56.78
**Fear of losing jobs**	30.72	20.55	48.73
**Fear of losing part of income**	22.88	20.76	56.36
**Fear of rising prices**	6.57	14.83	78.60
**Fear related to childcare provision**	75.64	11.65	12.71
**Fear of being quarantined**	31.36	27.97	40.68
**Fear of changing into remote working**	65.89	17.16	16.95
**Fear of ensuring working/learning conditions for household members**	63.98	16.74	19.28
**Fear of using public transport**	43.64	25.42	30.93
**Closing sports clubs**	55.30	23.09	21.61
**Closing catering facilities**	43.01	29.24	27.75
**Closing cultural institutions**	42.16	28.18	29.66
**Restrictions on movement and leaving apartments**	16.10	19.07	64.83
**Difficulties in accessing health services**	10.59	19.70	69.70
**Need to wear masks**	45.55	24.15	30.30
**Fear of ignoring restrictions by others**	27.75	22.67	49.58
**Travel difficulties**	21.61	24.15	54.24

Source: own study on the basis of questionnaire research.

**Table 5 ijerph-20-05832-t005:** Correlation between the impact of greenery on stress and the subjective level of and change in stress (Chi^2^ test results and *p*-value).

Impact of Greenery on Stress		Stress	
		Stress Level	Change in Stress Level
**Elements of the City’s Structure Reducing Stress**	Total	34.9779 (*p* = 0.00003)	22.5620 (*p* = 0.0002)
	Green spaces (parks, small green areas, allotments)	52.5365 (*p* = 0.00000001)	23.2391 (*p* = 0.0001)
	Outdoor recreational areas (outdoor gyms, sports fields, tennis courts, playgrounds, etc.)	29.1069 (*p* = 0.0003)	15.1410 (*p* = 0.0044)
**Ways of Reducing Stress**	Total	30.6368 (*p* = 0.0002)	11.2580 (*p* = 0.0238)
	Outdoor activity (walking, jogging, biking, etc.)	3.8982 (*p* = 0.4200)	5.7678 (*p* = 0.0559)
	Being surrounded by greenery (parks, gardens, small green areas)	25.4733 (*p* = 0.00004)	6.6933 (*p* = 0.0352)
	View of greenery/water/nature	13.3646 (*p* = 0.0096)	6.7633 (*p* = 0.0340)
	Garden work, plant cultivation	3.1214 (*p* = 0.5377)	4.4112 (*p* = 0.1102)

Source: own study on the basis of the questionnaire results.

## Data Availability

Data available upon request due to ethical restrictions. The data presented in this study are available upon request from the corresponding author. Data are not publicly available due to the use of only a portion of the data collected as part of the implementation of a broader project.

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
