# Peer review of "The Role of Greenery in Stress Reduction among City Residents during the COVID-19 Pandemic"

_ijerph, 2023, doi:10.3390/ijerph20105832_

Round 1
Reviewer 1 Report
The topic of this paper is of great current interest. Although it is focused on the city of Poznan, it draws interesting findings regarding the overall urban dwellers. The manuscript is well-written, but I raise some concerns about the structure: in my opinion, is not fluid, which difficult the reading and understanding of the study. Chapters 3 and 4 seem introductory since they present a literature review on the study's main topics: the city's stressors and the role of greenery in reducing stress. That is why I do not understand why this information comes after the methodology section. This knowledge seems to have been fundamental to formulate your questionnaire, but that relationship is not clear. I found these chapters very interesting to read (although rather long in the context of the manuscript) but I think the most important aspect should be to indicate how the information you reviewed shaped your questions. How and why did you select what to evaluate through your questionnaire? This is also unclear because the methodology section lacks much important information. At this stage, the reader cannot understand how the questionnaire is structured and what it evaluates clearly, and that is mandatory to understand the study in itself. What variables were selected? Did you use scales? How the selection of variables relates to your hypothesis and objectives? One gets a notion of that later in the paper, in the results sections, but I do not think is enough. This problem in the manuscript is very easy to resolve since you know precisely how the questionnaire was formulated and why. Also, you can include the questionnaire as supplementary material. Chapters 1, 3 and 4 can even be reduced to expand the methods (which should be restructured to come after what is now chapters 3 and 4). Another hypothesis is merging chapters 3 and 4 into the introduction, reducing the text and creating subchapters. This could also give you more space to expand the discussion of your results.
Specific comments:
Lines 65-67: This sentence seems incomplete. Maybe “Urgent improvement in the knowledge ... is needed.”??
Line 69: The “I” seems to be a typo.
Lines 81-86: The hypothesis and objectives of the study are quite challenging to follow in the present form, maybe because there are several objectives included. I would suggest rephrasing the paragraph to facilitate reading.
Line 82: How do you define greenery? And what are landscaped green spaces? Are you focused on public urban green spaces, or the study also includes private green spaces? It would improve the text if you could clarify this.
Lines 87-88: Could you give some examples of research that focus on recreational or ecological approaches?
Lines 102-105: Unclear. Can you specify which method was used for each objective? For instance, how did you determine the level and sources of city residents’ stress during the pandemic? Only through a geo-questionnaire? What kind of documents and source materials did you need? What may seem obvious for you, may not be to readers who are not aware of the study dimensions while reading it for the first time.
Line 106: What exactly is a geo-questionnaire? It may be useful to explain the difference compared to questionnaires.
Lines 113-119: I would suggest adding more information to this paragraph. It is not entirely clear whether the geo-questionnaire was only applied online or, for instance, how it was structured. How many questions does it include? How long it takes to answer? How did respondents receive the geo-questionnaire (through e-mail?)? Was it tested before application? etc... Another aspect that seems odd is that you are already giving result information here (e.g., gender, age, education, ...) Maybe there's a valid reason to do it, but what is that?
Line 116: Not sure "interviewees" is the correct word. They were not interviewed, but rather inquired. Maybe respondent is the fittest word.
Lines 124-141: Statistical analysis seems appropriate, and the description is helpful.
Table 2: Please include some gaps between the different lines of the table so it is easier to read which examples refer to each group of stressors. Also, how were these stressors analyzed through the questionnaire? This information should be in the method section, and a clear relationship between the questionnaire and the reviewed literature in chapters 3 and 4 should also be made. That is why I believe the structure of the manuscript should be revised.
Chapter 5: Some parts of the text belong in the method section, e.g., “respondents assessed their stress level on the 0–10 scale (0–lack of stress; 10–the highest stress level)”… and also the information about which stressors you evaluated.
Figure 4: This is a very interesting figure! I would only advise increasing the circles to improve reading or even trying more contrasting colours. I imagine you were only able to spot these places because you applied a geo-questionnaire. Before I read this manuscript, I was not aware of geo-questionnaires and I had to search about it because you do not explain it here. I would like to understand better how you constructed this map, and I think it would improve the manuscript if you included that information in the text.
Lines 382-384: This realization should have been at the beginning of the manuscript. This is why you had the need to include reviewed literature in chapters 3 and 4.
Line 404: Remove the full stop.
Lines 419-436: I think is very interesting to bring to this discussion the Biophilia concept, but I think is only strange that it only appears in the discussion section, if I’m not mistaken. Could it also be mentioned in the introduction?
Conclusions: The beginning of this chapter seems very repetitive, not adding much more than the discussion. The recommendations, on the other hand, are interesting and should be emphasized and more detailed. I would even recommend you create a chapter or subchapter dedicated to these recommendations that you extracted from the results of the study and transform the conclusion on something more concise that focuses on the main findings but also on the limitations of the study and future research (which are now missing from the text). I was expecting the recommendation to be more detailed regarding actions that need to be implemented in the city of Poznan, also because you have spatial information to do it (Figure 4). For instance, how and where would you increase the proportion of green space? Are there abandoned green areas in the city that could be transformed into parks? Or streets that could be revindicated only to pedestrians? How would you ensure that green space is in proximity to urban residents?
Lines 482-491: While interesting, this does not seem the best way to conclude your manuscript. You are stating the benefits of greenery that should be on chapter 4. Here you should be demonstrating why your findings are essential to improve the quality of life of urban dwellers and that more research like yours should be replicated in other urban contexts.
Reviewer 2 Report
Dear authors,
Thank you for the opportunity to review the paper "The role of greenery in building a healthy, stress-resilient city." As urban greenery is gaining renewed interest and popularity, it is essential to have case studies like this that demonstrate how green areas improve the resilience of cities and what factors of the urban green system contribute to their resilience. While the paper's topic is promising, consistent improvements and clarifications are needed to be publishable.
I appreciate the work authors have done in preparing the manuscript, but I think the manuscript is not in good shape yet to be ready for acceptance. Please reconsider how to convert your paper into an academic issue with sound knowledge gaps, aims, questions, and methods. Some parts are irrelevant (generic definition of stress and linked references, lines 143-171 of section 3; discussion regarding biophilic concept without explaining the links between these concepts and the authors' findings), and other parts need more elaboration (e.g., organization of the questionnaire template, data about the urban green system of the case study, assumptions and recommendations need to be made using the findings).
Successively I list my considerations.
Abstract. The authors have used literature analysis (unclear how) and questionnaires conducted among Poznań city residents to verify their hypothesis that greenery reduced stress among urban dwellers during the COVID-19 pandemic. Unfortunately, the abstract does not provide specific quantitative data on the number of interviews carried out or the percentage of participants who affirmed the resulting statements. However, the authors mention that the analysis revealed that the interviewees declared above-average stress levels during the pandemic, which increased due to the restrictions imposed. The authors also note that green areas and outdoor activities helped reduce this stress.
Additionally, the authors note that the respondents perceived a post-pandemic city as greener and that priority should be given to unmanaged green areas. The abstract provides a general overview of the study's findings and the research procedure adopted to verify the hypothesis. I suggest adding some specific quantitative data (e.g., the number of interviews carried out, the percentage of participants who affirmed the resulting statements, and the green areas per capita or other information about the amount of greenery in the study area )
1 Introduction
1.1) The authors need to clarify as they define the term "resilience." Indeed they affirm that "resilience can be understood as a product of the risk that a stressor will occur, vulnerability to it, which is the combination of susceptibility and exposure to stress as well as the duration of exposition to it" (lines 70-72). I suggest better explaining this key concept.
For an in-depth explanation of the meaning of resilience, authors could use these references:
- Jones, L., d’Errico, M., 2019. Whose resilience matters? Like-for-like comparison of objective and subjective evaluations of resilience. World Dev. 124 https://doi.org/ 10.1016/j.worlddev.2019.104632
The authors affirm that "resilience" describes how a system responds to shocks by adapting to changing circumstances while maintaining essentially the same functions and structures.
Menconi, M.E., Palazzoni, L., Grohmann, D. Core themes for an urban green systems thinker: A review of complexity management in provisioning cultural ecosystem services. Urban Forestry & Urban Greening, 2021, 65, 127355
Where authors develop a review regarding complex system approaches in greenery and, in section 4.5, they discuss how urban green planners and designers use the urban green system to improve the resilience of the city-system
1.2) The paper aims to verify whereby greenery reduced urban dwellers' stress during the pandemic. Furthermore, the authors aim to list the sources of stress, the role of vegetation in mitigating it, and to give an "indication of a spatial urban structure model conducive to improving the resilience of city dwellers to stress and building a stress-resilient city" (lines 85-86).
I suggest deleting the last aim because the authors did not develop an urban green planning method. Alternatively, the authors need to add quick literature research about the main criteria used in urban green planning (e.g., a. Biernacka, M.; Kronenberg, J. Classification of institutional barriers affecting the availability, accessibility, and attractiveness of urban green spaces. Urban For. Urban Green 2018, 36, 22–33. b. Biernacka, M.; Kronenberg, J.; Łaszkiewicz, E. An integrated system of monitoring the availability, accessibility and attractiveness of urban parks and green squares. Appl. Geogr. 2020, 116.; c. Menconi, M.E.; Sipone, A.; Grohmann, D. Complex Systems Thinking Approach to Urban Greenery to Provide Community-Tailored Solutions and Enhance the Provision of Cultural Ecosystem Services. Sustainability 2021, 13, 11787), and biophilic design (a. Edward Osborne Wilson, Biophilia, 1984, Harvard University Press, ISBN 0-674-07441-6. b. Stephen Robert Kellert, The Value of Life: Biological Diversity and Human Society, Stephen R. Kellert. 1996. Island Press, Washington, DC. 280 pages. ISBN: 1-55963-317-4, Volume 16, Issue 4. https://doi.org/10.1177/027046769601600415; c. their reference n. 72: Kellert, S. R.; Heerwagen, J. H.; Mador, M. L. Biophilic Design. The Theory, Science and Practice of Bringing Buildings to Life. 658 Wiley: Hoboken, USA, 2008).
Then, the authors should use this literature review in the discussion section to explain how their findings could help improve urban green system planning.
1.3) Authors should move the literature review of section 4 regarding the relationship between health and greenery to a subsection of the introduction
1.4) Finally, I suggest better clarify which is the gap in the current research.
2 Material and methods (called by authors: Research methods)
2.1) Line 108 – substitute the term Statistics Poland with a reference
2.2) The authors collected 651 responses to an online questionnaire. Which criteria to select the group target (e.g., living in the city chosen as the case study?)How are participants invited to respond to the questionnaire? How did the authors select the participants? Have they invited all the inhabitants of the city ? How?
2.3) I suggest adding in an appendix the questionnaire-template
3 Stress: Its sources and effects in cities
3.1) This section reports the literature analysis about stress and its sources.
I suggest deleting this section (too generic for the aim of the paper) or moving it to a subsection of the introduction because, at the moment, it is only a state-of-art (methods developing a literature review have protocols of review using clear rules of selection between papers and precise steps of the analysis).
3.2) Substitute references Ellison, Maynard, 1992 (line 149), and Hagger et al., 2020 (line 232) with the corresponding numbers [1] and [6]
4. The role of greenery in reducing urban dwellers' stress
4.1) I suggest moving this section to a subsection of the introduction
5. Questionnaire results
5.1) Without the questionnaire template, it is impossible to understand in-depth how the research was conducted, so analyzing the results is difficult. See other considerations for this section in the recommendations about tables and figures
6. Discussion of the results
6.1) In this section, authors should compare their results with others in the literature and support every statement with their previous findings.
(only one example, Lines 385-386 "The results of the research conducted among Poznań residents confirm the theses developed in the literature that urban dwellers perceive their stress level as above-average [1,2] and that the COVID-19 pandemic exacerbated this stress."
I think that authors should comment on this statement also using their findings in In lines 305-306 "Over 40% of the respondents declared an increase in the stress level experienced during the pandemic, whereas 35.81% noticed its decrease, and 23.52% of the respondents stated that stress was maintained at the pre-pandemic level."
The authors are sure that, using these results, they can affirm that the COVID-19 pandemic exacerbated citizens' stress. What about the 35.81% of the respondents that were more relaxed during the restriction of covid-19 pandemic?)
6.2) Biophilic is a significant concept that has guided urban green planners and designers, but the authors add trivial reflections about its principles without a clear link with their findings.7.
7. Conclusions
7.1) The recommendations need a critical reflection - in my opinion, many of them are not backed up by their research in this paper.
Revise all the recommendations in-depth and select only those linked with this paper's findings.
Only one example for recommendation n. 5 "It is important to enabling residents to interact with nature as often as possible daily (at least a five-minute contact with nature in a local living environment, both at the residence and workplace)."
Which findings of the authors allow them to make this statement? When have they studied the time spent in the green? Indeed, they also have not studied the role of green in the workplace during the quarantine.
Tables
T.1) Statements inside the text could substitute tables 2 and 3.
Figures
F.3) Figure 3 I suggest explaining outdoor and indoor categories. Indeed, observing the figure, the main factor reducing the stress seems to be the outdoor areas rather than the greenery. Why are green spaces in the first place? Because they are outdoor areas or because they have vegetation. Fortunately, the authors added figure 5, with significant findings for greenery. Anyway, the authors should add the questionnaire template to understand the results (e.g., in figure 5, the participants need to respond to an open or closed question. The categories reducing stress listed in figure 5 result from grouping between authors' open responses, or the participants must choose between closed categories)
F.4) The map of Figure 4 shows specific places within the city exacerbating or reducing the respondents' stress. I would like to know if the online questionnaire showed a map where citizens could add a waypoint. Otherwise, how have the authors found these points?
Reviewer 3 Report
This manuscript is very interesting and valuable, but I still have some problems as follows:
1. How to understand the stress in the cities in epistemology?
This manuscript aims to verify a hypothesis whereby greenery reduces the level of urban dwellers' stress during the pandemic.
But first of all, I confused about the reason why the citizens/urban dwellers feel stress, mainly is (1) the stress itself, anywhere in the cities, which is constantly existing; (2) the exacerbated stress from social isolation or economic effects during COVID-19. On the one hand, as this paper pointed “ stress is one of the main health challenges of the 21st century”. There is no downtime in tiered cities because of the febrile rhythm and hastening speed of today’s crisis-prone, globalized, financialised capitalism (Madden, 2022). On the other hand, the authors’ hypothesis that stress mainly results from social distancing and movement restrictions associated during COVID-19?
Secondly, what is the difference between stress and reducing stress before COVID-19 and after COVID-19? As the authors said in Line 27 said:” We all live in a world full of stress” , and the authors noted in Line 258-260, “Not only does staying in greenery reduce stress but also looking at it [56]. Therefore, contact with the natural environment is conducive to mental renewal [58], mood improvement [46,57,59], and also concentration improvement [46,47].”
I also believe it how stressed the citizens are in globalized, financialised, busy, and tired world-cities, and how comfortable the citizens are in green areas and outdoor activities, to be surrounded by nature and greenery can help the citizens reduce the stress, whatever time you are, before COVID-19, During COVID-19, or post-COVID-19.
So the author should tell the readers the differences between stress and reducing stress before COVID-19 and after COVID-19.
2.Some discourses should be more accurate.
Such as Line 54-55, “Recently, the COVID-19 pandemic has become an acute stressor for urban dwellers, leaving its particular mark just in cities.” As WHO reports, COVID-19 almost has been gone.
3.Repeated quotations marked with the page number of the bibliography or book or paper, such as [1], [2],[3], [4], etc. the author citied in Line 38, 44, 49,53, 49, 150,, 155, 168, 171, 177, 181,182, 265, etc., [2] Cited in Line 52, 65, 67, 70, 159, 177, 184, etc.
4. Some questions need to be need to be clarified in research method
Firstly, as the author said,"the most stressful element for the residents during the pandemic was not the virus itself, but a series of the resultant restrictions imposed in urban space. "
Line 388-289 said, "since March 2022, almost all pandemic constraints have been removed in Poland", but the geo-questionnaire studies were carried out in July–September 2022 rather than carried out during COVID-19.
Will this factor affect the stress feelings of the citizens?
Secondly, the author needs to clarify whether the respondents in this study were excluded. For example, some people like green and outdoor activities; others maybe rely on watching Netflix or Japanese comics or shopping to relieve stress.
5.As figure 4 shows, I have a question or, more likely, a suggestion, the distribution of pressure in geographical space is different, as we all know, uneven development, the inequality between urbanization and greening, and the inequality of social groups (the rich and white people live, or low-income families and working class), the authors could consider these factors in Conclusion if possible.
Madden, D. (2022). Tired city: on the politics of urban exhaustion. City, 26(4), 559-561.
Round 2
Reviewer 1 Report
I am pleased to see that the authors have made the solicited modifications to the manuscript, resulting in my opinion in a much-improved work. I have no further questions or recommendations.
Reviewer 2 Report
Dear authors,
congrats on the revised paper. You have revised your paper in-depth, and now, the aims of the research, the method, and the results are clear. At the same time, your revision highlights that the paper's title does not reflect its content, which mainly studies the relationship between pandemic stress and the city's spaces. The great value of the urban green system in reducing stress is your main finding but not the focus of this paper. Indeed, 4 goals of your research study the relationship between pandemic stress and the city's spaces, and only the last one refers explicitly to urban greenery. The structure of your questionnaire (supplementary material) and your results reflect these goals (there is no one table or figure dedicated to the role of the urban green system.)
I suggest changing the title of the paper to recall its main themes.
Furthermore, reading the questionnaire (supplementary material), It is unclear how the authors built Table 6. Indeed the first and the second categories ( "Green spaces (parks, small green areas, allotments" and "Outdoor recreational areas (outdoor gyms, sports fields, tennis courts, playgrounds, etc.") are two of the 16 categories that the authors have defined for the question "The use of which elements of the city structure increases the stress connected with the pandemic that you perceive/ feel, and which reduces it." At the same time, to respond to this question, authors did not use the category "Forest," so how do authors evaluate this third category in Table 6?
I suggest revising this table, adding the two questions (1. "The use of which elements of the city structure increases the stress connected with the pandemic that you perceive/ feel, and which reduces it" 2. "What helps you reduce the level of stress you perceive/ feel during the pandemic" ) and using all the categories of the questionnaire (16 categories for the first question and 15 for the second).
Generally, about these two questions, in the questionnaire, authors ask respondents simply a dichotomic value (yes or no), and there is no explanation about the selection of the used categories in every question (some of them are trivial).
For future papers, I suggest using more effective categories and Likert scales for the evaluations to achieve more fully sound findings.
